# COVID-19 vaccine hesitancy and confidence in the Philippines and Malaysia: A cross-sectional study of sociodemographic factors and digital health literacy

Ken Brackstone[1]*, Roy R. Marzo[2,3,4], Rafidah Bahari[5], Michael G. Head[1], Mark E. Patalinghug[6], Tin T. Su[4,7]

1 Clinical Informatics Research Unit, Faculty of Medicine, University of Southampton, Southampton, United Kingdom, 2 Department of Community Medicine, International Medical School, Management and Science University, Shah Alam, Malaysia, 3 Department of Community Medicine, Faculty of Medicine, Asia Metropolitan University, Johor Bahru, Malaysia, 4 Global Public Health, Jeffrey Cheah School of Medicine and Health Sciences, Monash University Malaysia, Bandar Sunway, Malaysia, 5 Department of Psychiatry, Faculty of Medicine, University of Cyberjaya, Cyberjaya, Malaysia, 6 Department: School of Criminal Justice Education, Institution: J.H. Cerilles State College, Caridad, Dumingag, Zamboanga del Sur, Philippines, 7 South East Asia Community Observatory (SEACO), Jeffrey Cheah School of Medicine and Health Sciences, Monash University Malaysia, Bandar Sunway, Malaysia

* K.Brackstone@soton.ac.uk

**Data Availability Statement:** Data are available on the OSF repository: https://osf.io/ncwjq/.

## Abstract

With the emergence of the highly transmissible Omicron variant, large-scale vaccination coverage is crucial to the national and global pandemic response, especially in populous Southeast Asian countries such as the Philippines and Malaysia where new information is often received digitally. The main aims of this research were to determine levels of hesitancy and confidence in COVID-19 vaccines among general adults in the Philippines and Malaysia, and to identify individual, behavioural, or environmental predictors significantly associated with these outcomes. Data from an internet-based cross-sectional survey of 2558 participants from the Philippines ($N$ = 1002) and Malaysia ($N$ = 1556) were analysed. Results showed that Filipino (56.6%) participants exhibited higher COVID-19 hesitancy than Malaysians (22.9%; $p$ < 0.001). However, there were no significant differences in ratings of confidence between Filipino (45.9%) and Malaysian (49.2%) participants ($p$ = 0.105). Predictors associated with vaccine hesitancy among Filipino participants included women (OR, 1.50, 95% CI, 1.03–1.83; $p$ = 0.030) and rural dwellers (OR, 1.44, 95% CI, 1.07–1.94; $p$ = 0.016). Among Malaysian participants, vaccine hesitancy was associated with women (OR, 1.50, 95% CI, 1.14–1.99; $p$ = 0.004), social media use (OR, 11.76, 95% CI, 5.71–24.19; $p$ < 0.001), and online information-seeking behaviours (OR, 2.48, 95% CI, 1.72–3.58; $p$ < 0.001). Predictors associated with vaccine confidence among Filipino participants included subjective social status (OR, 1.13, 95% CI, 1.54–1.22; $p$ < 0.001), whereas vaccine confidence among Malaysian participants was associated with higher education (OR, 1.30, 95% CI, 1.03–1.66; $p$ < 0.028) and negatively associated with rural dwellers (OR, 0.64, 95% CI, 0.47–0.87; $p$ = 0.005) and online information-seeking behaviours (OR, 0.42, 95% CI, 0.31–0.57; $p$ < 0.001). Efforts should focus on creating effective interventions to decrease

**Funding:** The authors received no specific funding for this work.

**Competing interests:** The authors have declared that no competing interests exist.

vaccination hesitancy, increase confidence, and bolster the uptake of COVID-19 vaccination, particularly in light of the Dengvaxia crisis in the Philippines.

## Introduction

While many high-income settings have achieved relatively high coverage with their COVID-19 vaccination campaigns, almost 32.1% of the world's population have not received a single dose of any COVID-19 vaccine as of July 2022 [1]. The Philippines and Malaysia are among two of the most populous countries in Southeast Asia with an estimated population of 110 million and 32 million people, respectively. To date, Malaysia has seen over 4.6 million cases with a mortality rate of 0.77%, while approximately 3.7 million cases of COVID-19 were detected in the Philippines with a mortality rate of 1.60% [2]. Malaysia is doing considerably well with their vaccination efforts, with 84.8% of the population currently considered fully vaccinated as of July 2022. However, vaccination campaigns in the Philippines have been more difficult, with 65.6% of the population fully vaccinated [3]. With the emergence of the highly transmissible Omicron variant across the world [4], large-scale vaccination coverage remains fundamental to the national and global pandemic response. Regular scientific assessments of factors that may impede the success of COVID-19 vaccination coverage will be critical as vaccination campaigns continue in these nations.

A key factor for the success of vaccination campaigns is people's willingness to be vaccinated once doses become accessible to them personally. Vaccine hesitancy is defined by the World Health Organization (WHO) as the delay in the acceptance, or blunt refusal of, vaccines. In fact, vaccine hesitancy was described by the WHO as one of the top 10 threats to global health in 2019 [5]. Conversely, vaccine confidence relates to individuals' beliefs that vaccines are effective and safe. In general, a loss of trust in health authorities is a key determinant of vaccine confidence, with misconceptions about vaccine safety being among the most common reasons for low confidence in vaccines [6].

Previously, vaccination in Southeast Asia has been associated with mistrust and fear, particularly in the Philippines, who are still suffering the consequences of the Dengvaxia (dengue) vaccine controversy in 2017 [7]. Studies suggest that this highly political mainstream event, in which anti-vaccination campaigns linked dengue vaccines with autism spectrum disorder and with corrupt schemes of pharmaceutical companies, continue to erode the population's trust in vaccines. For example, a survey conducted on over 30,000 Filipinos in early 2021 showed that 41% of respondents would refuse the COVID-19 vaccine once it became available, whereas Malaysia reported 27% hesitancy [8]. Researchers predict that the controversy surrounding Dengvaxia may have prompted severe medical mistrust and subsequently weakened the public's attitudes toward vaccines [7, 9]. However, there may be many additional factors that weaken confidence in vaccines. For example, incompatibility with religious beliefs is one key driver of weakened confidence in vaccines [10, 11], whereas living in urbanised (vs. rural) areas predicts COVID-19 vaccine hesitancy in some countries [12–14], possibly due to being more connected to the internet and social media and being more exposed to COVID-19-related misinformation.

Other predictors of vaccine hesitancy and confidence may include digital health literacy–one's ability to seek, find, understand, and appraise health information from digital resources–and social media use. Research has shown that beliefs in available information is integral to perceptions of the vaccine safety and effectiveness [15–17]. Previous studies, for example, have

associated higher vaccine hesitancy with misinformation about the virus and vaccines, particularly if they relied on social media as a key source of information [18, 19]. Social Cognitive Theory (SCT) is a widely accepted theory which may explain individual behaviors, including digital health literacy [20]. SCT consists of three factors–environmental, personal, and behavioural–and any two of these components interact with each other and influence the third. As such, SCT can assist in establishing a link between one's behaviour (e.g., information-seeking–one form of digital health literacy) and environmental factors (e.g., availability of information online), which may interact to promote medical mistrust and influence vaccine hesitancy and confidence (personal) [21]. Thus, health behaviours are often influenced by social systems as well as personal behaviours.

Although vaccine hesitancy and confidence are related concepts (e.g., people who express low confidence in vaccines are more likely to be vaccine-hesitant [6]), they are also distinct [22]. Thus, the main aims of this research were to determine levels of hesitancy and confidence in COVID-19 vaccines among general adults in the Philippines and Malaysia, and to identify behavioural or environmental predictors that are significantly associated with both outcomes. Thus, developing a deeper understanding of the factors associated with vaccine hesitancy and confidence will provide insight into how specific population groups may respond to health threats and public health control measures.

## Methods

### Design, subjects, and procedure

This was an internet-based cross-sectional survey conducted from May 2021 to September 2021 in the Philippines and Malaysia. Snowball sampling methods were used for the data collection using social media, including research networks of universities, hospitals, friends, and relatives. Filipino and Malaysian residents aged 18 years or older were invited to take part. The inclusion criteria for participants' eligibility included 18 years or older, and an understanding of the English language. All invited participants consented to the online survey before completion. Consented participants could only respond to questions once using a single account. The voluntary survey contained a series of questions which assessed sociodemographic variables, social media use, digital literacy skills in health, and attitudes toward the COVID-19 vaccine.

### Ethical approval

The study received ethical approval from Asia Metropolitan University's Medical Research and Ethics Committee (Ref: AMU/FOM/MREC 0320210018). All participants provided informed consent. All study information was written and provided on the first page of the online questionnaire, and participants indicated consent by selecting the agreement box and proceeding to the survey.

### Measures

**Demographics.** Filipino and Malaysian participants indicated their age category (18–24, 25–34, or 35–44), gender (man, woman), community type (rural, urban), educational level (no formal education, primary, secondary, tertiary), employment (unemployed, part-time, full-time), religion (Christian, Buddhism, Muslim, Hinduism, Other, None), income (1 = *very insufficient*; 4 = *very sufficient*; $M$ = 1.84, $SD$ = 0.81), whether they were permanently impaired by a health problem (no vs. yes), and whether they were social media users (no vs. yes).

**Subjective social status.** Participant then rated their own perceived social status using the MacArthur Scale of Subjective Social Status scale [23]. Participants viewed a drawing of a

ladder with 10 rungs, and read that the ladder represented where people stand in society. They read that the top of the ladder consists of people who are best off, have the most money, highest education, and best jobs, and those at the bottom of the ladder consists of people who are worst off, have the least money, lowest education, and worst or no jobs. Using a validated single-item measure, participants placed an 'X' on the rung that best represented where they think they stood on the ladder (1 = *lowest*; 10 = *highest*; *M* = 6.23, *SD* = 1.86).

**Vaccine confidence and hesitancy.** Participants were also asked about their perceived level of confidence in the COVID-19 vaccine ("I am completely confident that the COVID-19 vaccine is safe," 1 = *strongly disagree*; 7 = *strongly agree; M* = 4.57, *SD* = 1.48). Then, participants were asked about their level of hesitancy to the COVID-19 vaccine ("I think everyone should be vaccinated according to the national vaccination schedule"; no, I don't know, yes). These questions were adapted from the World Health Organization, Regional Office for Europe survey [24]. The tool underwent evaluation by multidisciplinary panel of experts for necessity, clarity, and relevance.

**Digital health literacy.** Finally, participants completed the Digital Health Literacy Instrument (DHLI) [25], which was adapted in the context of the COVID-HL Network. The scale measures one's ability to seek, find, understand, and appraise health information from digital resources. A total of 12 items (three per each dimension) were asked, and answers were recorded on a four-point Likert scale (1 = *very difficult*; 4 = *very easy;* α = .92; *M* = 2.15, *SD* = 0.59). While the original DHLI is comprised of 7 subscales, we used the following four domains, including: (1) information searching or using appropriate strategies to look for information (e.g., "When you search the internet for information on coronavirus virus or related topics, how easy or difficult is it for you to find the exact information you are looking for?"; α = .87; *M* = 2.15, *SD* = 0.65), (2) adding self-generated content to online-based platforms (e.g., "When typing a message on a forum or social media such as Facebook or Twitter about the coronavirus a related topic, how easy or difficult is it for you to express your opinion, thought, or feelings in writing?"; α = .74; *M* = 2.15, *SD* = 0.65), (3) evaluating reliability of online information (e.g., "When you search the internet for information on the coronavirus or related topics, how easy or difficult is it for you to decide whether the information is reliable or not?"; α = .86; *M* = 2.20, *SD* = 0.69), and (4) determining relevance of online information (e.g., "When you search the internet for information on the coronavirus or related topics, how easy or difficult is it for you to use the information you found to make decisions about your health [e.g., protective measures, hygiene regulations, transmission routes, risks and their prevention?"]; α = .87; *M* = 2.09, *SD* = 0.68). The reliability statistics for the overall DHL score was 0.92, while the alpha coefficients for the four subscales ranged from 0.74 to 0.87, suggesting acceptable to good internal consistency.

## Data analysis

Data were examined for errors, cleaned, and exported into IBM SPSS Statistics 28 for further analysis. All hypotheses were tested at a significance level of 0.05. $\chi^2$ tests were conducted for group differences of categorical variables, and Mann-Whitney tests for continuous variables. Subgroup analyses were performed for Filipino and Malaysian participants.

COVID-19 vaccine hesitancy and confidence were treated as separate dependent variables in a logistic regression model providing the strictest test of potential associations with COVID-19 vaccine hesitancy and confidence among Filipino and Malaysian participants. Low vaccine confidence was operationalised by dichotomising participants' responses to the statement: "I am completely confident that the COVID-19 vaccine is safe" into those who disagreed or neither agreed nor disagreed (1–4), whereas high vaccine confidence was operationalised by

dichotomising participants' responses into those who agreed to some extent (5–7). Vaccine hesitancy was operationalised by dichotomising responses to the statement: "I think everyone should be vaccinated according to the National vaccination schedule" into those indicating 'no' or 'I don't know,' whereas no vaccine hesitancy was operationalized by dichotomising participants' response into those who indicated 'yes.'

Independent variables were: age (18–24 vs. 25–34 vs. 35–44 [ref]), gender (women vs. men [ref]), community type (rural vs. urban [ref]), educational level (tertiary vs. secondary or less [ref]), employment (employed to some degree vs. unemployed [ref]), religion (Philippines: Christianity vs. Islam [ref]; Malaysia: Christianity vs. Buddhism vs. Hinduism vs. Islam [ref]), income (low (1–2) vs. high (3–4 [ref])), whether they were permanently impaired by a health problem (yes vs. no [ref]), whether they were social media users [yes vs. no [ref]), their perceived ranking on the MacArthur Scale of Subjective Social Status (continuous variable), and finally the four domains of the DHLI scale (all continuous variables).

## Results

A total of 2558 participants completed the online survey. Table 1 shows descriptive statistics of participants from the Philippines ($N$ = 1002) vs. Malaysia ($N$ = 1556). Filipino (vs. Malaysian) participants indicated higher rates of education ($p < 0.001$), but were more likely to be unemployed ($p < 0.001$). Further, Filipino (vs. Malaysian) participants were also more likely to indicate lower income ($p < 0.001$) and rate themselves lower on subjective social status ($p < 0.001$). Malaysian (vs. Filipino) participants were more likely to live in urban areas ($p < 0.001$). Most notably, Filipino participants (56.6%) indicated higher prevalence of COVID-19 vaccine hesitancy compared to Malaysian participants (22.9%; $p < 0.001$). However, there were no significant differences between Filipino (45.9%) and Malaysian (49.2%) participants in ratings of vaccine confidence ($p = 0.105$). Malaysian (vs. Filipino) participants were also more likely to report using social media (96.6 vs. 89.8%; $< 0.001$).

Table 2 shows significant predictors of vaccine hesitancy in both Filipino and Malaysian samples. Among Filipino participants, multivariate logistic regression analyses revealed that factors associated with higher vaccine hesitancy included women (OR, 1.51, 95% CI, 1.14–2.00; $p = 0.004$), residing in a rural community (OR, 1.45, 95% CI, 1.07–1.95; $p = 0.015$), and having lower income (OR, 1.62, 95% CI, 1.20–2.19; $p = 0.001$). Among Malaysian participants, women (OR, 1.51, 95% CI, 1.14–2.00; $p = 0.004$), being aged 25–34 (vs. 18–24; OR, 1.52, 95% CI, 1.48–2.21; $p = 0.027$), Christians (OR, 2.45, 95% CI, 1.66–3.62; $p < 0.001$), completing tertiary education (OR, 2.17, 95% CI, 1.63–2.88; $p < 0.001$), social media use (OR, 11.59, 95% CI, 5.63–23.84; $p < 0.001$), and information-seeking behaviours (OR, 2.50, 95% CI, 1.74–3.61; $p < 0.001$) were predictors of higher vaccine hesitancy, whereas having a health impairment (OR, 0.49, 95% CI, 0.30–0.78; $p = 0.003$) and higher self-reported ratings on subjective social status (OR, 0.82, 95% CI, 0.75–0.89; $p < 0.001$) were associated with lower vaccine hesitancy.

Table 3 shows significant predictors of vaccine confidence in both Filipino and Malaysian samples. Factors positively associated with higher vaccine confidence among Filipino participants included higher self-reported ratings on subjective social status (OR, 1.16, 95% CI, 1.07–1.25; $p < 0.001$), whereas factors associated with lower vaccine confidence included women (OR, 0.72, 95% CI, 0.54–0.96; $p = 0.026$) and information-seeking behaviours (OR, 0.63, 95% CI, 0.49–0.81; $p < 0.001$). Among Malaysian participants, factors positively associated with higher vaccine confidence included women (OR, 1.27, 95% CI, 1.18–1.60; $p = 0.035$), completing tertiary education (OR, 1.31, 95% CI, 1.03–1.66; $p = 0.026$), and higher self-reported ratings on subjective social status (OR, 1.08, 95% CI, 1.00–1.16; $p = 0.036$). Factors negatively associated with lower vaccine confidence included residing in a rural community (OR, 0.63, 95% CI,

**Table 1. Socio-demographic characteristics of participants from the Philippines vs. Malaysia.** Values are presented as percent (n) or means ± SD.

| Characteristics | Total (N = 2558) | Philippines (N = 1002) | Malaysia (N = 1556) | *p* |
|---|---|---|---|---|
| **Gender** | | | | < 0.001 |
| Women | 54.1 (1384) | 60.8 (608) | 49.8 (775) | |
| Men | 45.9 (1174) | 39.2 (393) | 50.2 (581) | |
| **Age** | | | | < 0.001 |
| 18–24 | 72.2 (1846) | 84.3 (845) | 64.3 (1001) | |
| 25–34 | 19.1 (488) | 12.2 (122) | 23.5 (266) | |
| 35–44+ | 8.7 (224) | 3.5 (35) | 12.2 (189) | |
| **Community type** | | | | < 0.001 |
| Rural | 37.5 (958) | 66.3 (664) | 18.9 (294) | |
| Urban | 62.5 (1600) | 33.7 (338) | 81.1 (1262) | |
| **Highest level of education** | | | | < 0.001 |
| Senior secondary or lower | 40.2 (1029) | 30.7 (308) | 46.3 (721) | |
| Tertiary | 59.8 (1529) | 69.3 (394) | 53.7 (835) | |
| **Employment** | | | | < 0.001 |
| Unemployed | 72.9 (1864) | 81.4 (826) | 67.4 (1048) | |
| Employed or self-employed | 27.1 (694) | 18.6 (186) | 32.6 (508) | |
| **Religion** | | | | < 0.001 |
| Islam | 35.3 (904) | 2.4 (24) | 57.0 (880) | |
| Christian | 43.3 (1108) | 92.7 (929) | 11.6 (179) | |
| Buddhism | 9.1 (233) | 0.1 (1) | 15.0 (232) | |
| Hinduism | 11.7 (300) | 4.8 (48) | 16.3 (252) | |
| **Income** | | | | < 0.001 |
| Higher | 72.9 (1865) | 62.3 (624) | 79.8 (1241) | |
| Lower | 27.1 (693) | 37.7 (378) | 20.2 (315) | |
| **Subjective social status** | 6.23 ± 1.85 | 5.89 ± 1.92 | 6.45 ± 1.78 | < 0.001 |
| **Health impairment** | | | | < 0.001 |
| No | 85.5 (2186) | 80.6 (808) | 88.6 (1378) | |
| Yes | 14.5 (372) | 19.4 (194) | 11.4 (178) | |
| **Vaccine hesitancy** | | | | < 0.001 |
| Low | 63.9 (1634) | 43.4 (435) | 77.1 (1199) | |
| High | 36.2 (954) | 56.6 (567) | 22.9 (357) | |
| **Vaccine confidence** | | | | < 0.105 |
| Low | 52.1 (1332) | 54.1 (542) | 50.8 (790) | |
| High | 47.9 (1226) | 45.9 (460) | 49.2 (766) | |
| **Social media use** | | | | < 0.001 |
| No | 6.04 (155) | 10.2 (102) | 3.4 (53) | |
| Yes | 93.6 (2403) | 89.8 (900) | 96.6 (1503) | |
| **Digital Health Literacy** | | | | |
| Information-seeking | 2.15 ± 0.65 | 2.29 ± 0.57 | 2.07 ± 0.68 | < 0.001 |
| Self-generated content | 2.15 ± 0.65 | 2.29 ± 0.55 | 2.07 ± 0.69 | < 0.001 |
| Evaluating reliability | 2.20 ± 0.70 | 2.37 ± 0.57 | 2.11 ± 0.74 | < 0.001 |
| Determining relevance | 2.09 ± 0.68 | 2.22 ± 0.57 | 2.02 ± 0.73 | < 0.001 |

0.47–0.87; *p* = 0.004), Christians (OR, 0.50, 95% CI, 1.20–2.24; *p* < 0.001), Buddhists (OR, 0.15., 95% CI, 0.10–0.22; *p* < 0.001), Hindus (OR, 0.24., 95% CI, 0.17–0.34; *p* = 0.004), information-seeking behaviours (OR, 0.42, 95% CI, 0.31–0.58; *p* < 0.001), and determining relevance of online information (OR, 0.68, 95% CI, 0.51–0.92; *p* = 0.013).

**Table 2. Multivariate conditional logistic regression analysis of vaccine hesitancy in the Philippines and Malaysia.**

| Variables | OR | 95% CI | p-value |
|---|---|---|---|
| **The Philippines** | | | |
| Women | 1.509 | 1.031–1.832 | .030 |
| Rural community | 1.443 | 1.070–1.947 | .016 |
| Lower income | 1.616 | 1.199–2.179 | .002 |
| **R²** | | | **0.061** |
| **Malaysia** | | | |
| Women | 1.509 | 1.141–1.994 | 0.004 |
| 25–34 | 1.533 | 1.055–2.225 | 0.025 |
| Christian beliefs [ref: Islam] | 2.433 | 1.648–3.593 | < 0.001 |
| Higher/tertiary education | 2.172 | 1.634–2.887 | < 0.001 |
| Health impairment | 0.537 | 0.339–0.851 | 0.008 |
| Subjective social status | 0.827 | 0.759–0.902 | < 0.001 |
| Social media use | 11.763 | 5.718–24.198 | < 0.001 |
| Online information-seeking | 2.486 | 1.727–3.580 | < 0.001 |
| **R²** | | | **0.278** |
| **Total** | | | |
| Women | 1.421 | 1.169–1.728 | < 0.001 |
| Urban community | 1.524 | 1.235–1.881 | < 0.001 |
| Christian beliefs [ref: Islam] | 3.527 | 2.806–4.433 | < 0.001 |
| Hindu [ref: Islam] | 0.637 | 0.412–0.983 | 0.042 |
| Higher education | 1.524 | 1.249–1.861 | < 0.001 |
| Lower income | 1.422 | 1.146–1.764 | < 0.01 |
| Health impairment | 0.746 | 0.568–0.980 | 0.035 |
| Social media use | 7.138 | 3.939–12.935 | < 0.001 |
| Subjective social status | 0.895 | 0.847–0.945 | < 0.001 |
| Online information-seeking | 1.725 | 1.329–2.239 | < 0.001 |
| **R²** | | | **0.270** |

OR, odds ratio; CI, confidence interval. Multivariate logistic regression analyses using stepwise variable selection.

## Discussion

Malaysia and the Philippines are among the most populous countries in Southeast Asia. While the economic impact of the COVID-19 pandemic has been permanent in the Philippines, it has been shown thus far to be temporary in Malaysia [26]. Between January and October 2020, around 30,000 Malaysians had been infected by the virus with a mortality rate of 0.79%, while approximately 380,000 cases of COVID-19 were detected in the Philippines with a mortality rate of 1.9% [2]. Further, 61.8% of Malaysians had completed their vaccination up until September 2021, while the percentage of completed vaccinations during the same period in the Philippines was only 19.2% [27]. Vaccine uptake is likely to be a key determining factor in the outcome of a pandemic. Knowledge around factors which predict vaccine hesitancy and confidence is of the utmost important in order to improve vaccination rates. Thus, the core aims of this research were to determine levels of hesitancy and confidence in COVID-19 vaccines among general adults in the Philippines and Malaysia, and to identify behavioural or environmental predictors that are significantly associated with these outcomes.

**Table 3. Multivariate conditional logistic regression analysis of vaccine confidence in the Philippines and Malaysia–significant associations with confidence.**

| Variables | OR | 95% CI | p-value |
|---|---|---|---|
| **The Philippines** | | | |
| Subjective social status | 1.136 | 1.054–1.225 | < 0.001 |
| | | | |
| **R²** | | | **0.075** |
| **Malaysia** | | | |
| Women | 1.276 | 1.016–1.604 | 0.036 |
| Rural community | 0.646 | 0.476–0.875 | 0.005 |
| Christian beliefs [ref: Islam] | 0.503 | 0.354–0.715 | < 0.001 |
| Buddhist beliefs [ref: Islam] | 0.153 | 0.105–0.222 | < 0.001 |
| Hindu beliefs [ref: Islam] | 0.245 | 0.174–0.345 | < .001 |
| Higher/tertiary education | 1.309 | 1.030–1.662 | 0.028 |
| Subjective social status | 1.083 | 1.007–1.165 | 0.031 |
| Online information-seeking | 0.423 | 0.311–0.575 | < 0.001 |
| Determining relevance of information | 0.681 | 0.507–0.916 | 0.011 |
| **R²** | | | **0.257** |
| **Total** | | | |
| Christian beliefs [ref: Islam] | 0.591 | 0.481–0.727 | < 0.001 |
| Buddhist beliefs [ref: Islam] | 0.152 | 0.106–0.218 | < 0.001 |
| Hindu beliefs [ref: Islam] | 0.235 | 0.169–0.326 | < 0.001 |
| | | | |
| Subjective social status | 1.116 | 1.060–1.174 | < 0.001 |
| Online information-seeking | 0.517 | 0.407–0.657 | < 0.001 |
| **R²** | | | **0.175** |

OR, odds ratio; CI, confidence interval. Multivariate logistic regression analyses using stepwise variable selection.

First, while there were no significant differences in ratings of confidence in the COVID-19 vaccine between Filipino and Malaysian participants, Filipino (compared to Malaysian) participants expressed greater vaccine hesitancy. This may be a consequence of previous vaccine scares in the years leading up to the pandemic, including the Dengvaxia controversy in 2016 [7, 9]. Systematic reviews demonstrated that, by the end of 2020, the highest vaccine acceptance was in China, Malaysia, and Indonesia [28, 29]. The authors postulated that this elevated awareness was due to being among the first countries affected by the virus, hence resulting in greater confidence in vaccines [28].

Next, this study shows that women expressed greater vaccine hesitancy in both countries. The evidence base shows mixed findings, with other studies reporting higher hesitancy in women [30] or in men [31]. In some countries, the gender gap is not as substantial as others. In a large global study conducted in countries such as Russia and the United States, it was found that there is greater gender gap in vaccine hesitancy among men and women compared to countries such as Nepal and Sierra Leone [32, 33]. Unsurprisingly, what drives this hesitancy is the inclusion of pregnant women, where studies have consistently demonstrated that this population is more hesitant toward vaccination due to concerns for their babies [34]. Hence, after taking all consideration into account, gender differences in vaccine hesitancy cannot be supported with certainty. This also emphasises the need for tailored health promotion towards the key populations at risk.

There are clear differences in predictors of vaccine hesitancy in the Philippines and Malaysia. However, when results for both countries were combined, women, urban dwellers, those

of Christian faith, those with higher educational attainment, higher self-reported social class, social media use, and information-seeking tendencies remained as predictors of hesitancy. Urban-dwellers and individuals with more years of education have previously been demonstrated as predictors for vaccine hesitancy [35], but contradictory results have also previously been shown [36, 37]. Urban residents are typically more connected to the internet and social media and, thus, may be more exposed to vaccine-related misinformation than rural inhabitants who have fewer sources of information available to them [12–14]. Nevertheless, reports have shown higher vaccine refusals among those with strong religious beliefs such as the Amish Community in the United States and the Orthodox Protestants in the Netherlands [38], as well as some Muslim groups in Pakistan [18].

Frequent social media use is the only strong predictor for vaccine hesitancy in this study, followed by information-seeking behaviours. Research has identified that the safety and effectiveness of the vaccine is the primary concern that people have, including beliefs in available information [15–17]. Unfortunately, high internet literacy is a double-edged sword, since participants in this study preferred to seek information through social media, and thus may have been exposed to inaccurate information regarding COVID-19 vaccine. Previous studies have associated higher vaccine hesitancy with misinformation about the virus and vaccines [18], particularly if they relied heavily on social media as a key source of vaccine-related information [19]. A 2022 systematic review discovered that high social media use is the main driver of vaccine hesitancy across all countries around the globe, and is especially prominent in Asia [39]. Furthermore, vaccine acceptance and uptake improved among those who obtained their information from healthcare providers compared to relatives or the internet [40].

In terms of vaccine confidence, our findings show that those with higher subjective social status have higher confidence in vaccination, consistent with previous studies describing how those with a higher income had expressed willingness to pay for their COVID-19 vaccination if necessary [32, 41, 42]. Further, those of Christian, Buddhist, and Hindu faiths, as well as those with a tendency to seek out information, were associated with lower vaccine confidence. This is in keeping with the previous findings demonstrating that strong religious convictions are often tied to mistrust of authorities and beliefs about the cause of the COVID-19 pandemic, which is fuelled by social media [43]. Furthermore, concern on the permissibility of these vaccines in their religion reduces its acceptability [10]. However, it is interesting to note that, while the majority in Malaysia are Muslims, it did not reduce the rate of vaccine acceptance and confidence in the country.

These findings have important implications for health authorities and governments in areas focusing on improving vaccination uptake. Misinformation about vaccination greatly hampers vaccination efforts. Thus, not only is it important to understand how specific population groups are influenced by digital platforms such as social media, but it is imperative to provide the right information driven by governmental and non-governmental organisations [39]. This could be achieved by having community-specific public education and role modelling from local health and public officials, which has been shown to increase public trust [44]. Since the primary reason for hesitancy is concern about the safety of vaccines, it is crucial that education programmes stress the effectiveness and importance of COVID-19 vaccinations [45]. Participants in this study coped with the pandemic by seeking out new information, but they sought information from social media when information from the authorities was lacking or were viewed as untrustworthy, which may have contained erroneous information. One way to deter this is to empower information-technology companies to monitor vaccine-related materials on social media, remove false information, and create correct and responsible content [44].

Furthermore, behavioural change techniques have been found to be useful in stressing the consequences of rejecting the vaccine on physical and mental health [46]. The most effective

"nudging" interventions included offering incentives for parents and healthcare workers, providing salient information, and employing trusted figures to deliver this information [47]. Finally, since religious concerns have been prominent in reducing vaccine confidence and increasing hesitancy in this study, it is important to tailor messages to include information related to religion, and the use of religious leaders to spread these messages [48]. These are all important factors for increasing uptake of the COVID-19 vaccine, but also may be relevant in acceptability of routine immunisations as countries look to transition towards a post-pandemic delivery of healthcare.

A limitation of this study includes its cross-sectional design and the heterogeneity among participants, which meant that temporal changes in attitudes toward COVID-19 vaccines across time were not captured. Further, the need for internet access among Filipino and Malaysian participants limited the representativeness of the sample population. Thus, certain demographic were under-represented, including Filipino and Malaysian individuals over the age of 45, and people of lower socio-economic status. The surveys were also implemented in English, which may have limited the participation of target participants who were not fluent in English. In addition, due to space limitations, vaccine hesitancy and confidence were each captured using one item, which raises concerns of the items' validity and reliability. Finally, not all independent variables were accounted for, including medical mistrust [49], vaccine knowledge [50], and specific social media platforms used [11]. We also did not assess whether participants had received any doses of the COVID-19 vaccine previously. Future research should include more important predictors to build a broader picture of vaccine-related hesitancy and confidence in the Philippines and Malaysia, and more items should be utilised to tap into these concepts more comprehensively. Despite these limitations, the core strength of this study relates to its relatively large number of participants from both countries, and its comprehensive analysis of predictors to provide as a starting point going forward.

## Conclusions

The main aims of this research were to determine levels of hesitancy and confidence in COVID-19 vaccines among unvaccinated individuals in the Philippines and Malaysia, and to identify predictors significantly associated with these outcomes. Predictors of vaccine hesitancy in this study included the use of social media, information-seeking, and Christianity. Higher socioeconomic status positively predicted vaccine confidence. However, being Christian, Buddhist or Hindu, and the tendency to seek information online, were predictors of hesitancy. Efforts to improve uptake of COVID-19 vaccination must be centred upon providing accurate information to specific communities using local authorities, health services and other locally-trusted voices (such as religious leaders), and for the masses through social media. Further studies should focus on the development of locally-tailored health promotion strategies to improve vaccination confidence and increase the uptake of vaccination–especially in light of the Dengvaxia crisis in the Philippines.

## Supporting information

**S1 File. Inclusivity in global research questionnaire.**
(DOCX)

## Author Contributions

**Conceptualization:** Ken Brackstone, Rafidah Bahari, Michael G. Head, Mark E. Patalinghug, Tin T. Su.

**Data curation:** Ken Brackstone, Roy R. Marzo, Rafidah Bahari, Mark E. Patalinghug, Tin T. Su.

**Formal analysis:** Ken Brackstone.

**Methodology:** Roy R. Marzo.

**Project administration:** Roy R. Marzo.

**Resources:** Roy R. Marzo.

**Supervision:** Tin T. Su.

**Validation:** Roy R. Marzo, Rafidah Bahari, Mark E. Patalinghug, Tin T. Su.

**Writing – original draft:** Ken Brackstone, Roy R. Marzo, Rafidah Bahari, Michael G. Head, Mark E. Patalinghug, Tin T. Su.

**Writing – review & editing:** Ken Brackstone, Roy R. Marzo, Rafidah Bahari, Michael G. Head, Mark E. Patalinghug, Tin T. Su.

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
