## [Decision Letter · Decision Letter 0]

12 Jul 2022

PGPH-D-22-00972

COVID-19 Vaccine Hesitancy and Confidence in the Philippines and Malaysia: A Cross-sectional Study of Sociodemographic Factors and Information-Seeking

Dear Dr. Brackstone

Thank you for submitting your manuscript to PLOS Global Public Health. After careful consideration, we feel that it has merit but does not fully meet PLOS Global Public Health’s publication criteria as it currently stands. Therefore, we invite you to submit a revised version of the manuscript that addresses the points raised during the review process.

EDITOR'COMMENTS: The reviewers agree that the paper is important as it addresses an important contemporary issues regarding population health for this time. Therefore I would encourage you to address these major issues they have pointed out, as summarized here but you can find their detailed comment attached;

Kindly take another look at the title of the study, it requires minor revision,Review the structure and coherence of the abstract and include aspects that are missing as mentioned by the reviewers,The two reviewers have pointed out theoretical and conceptual issues requiring clarifications, kindly address these issues as it would strengthen the manuscript the more and actually facilitate elucidation of the dynamics of the problem phenomenon.Note that strong review of theoretical framework provides two significant contributions; 1) provides scientific basis for the argument about the dynamics of the problem phenomenon with its justifications, and 2) the validity of data collected and conclusions contribute to strengthening the theories applied as basis for the study thesis.Kindly also address the referencing issues raised.

We look forward to receiving your revised manuscript.

Kind regards,

Nnodimele Onuigbo Atulomah, PhD

Academic Editor

Journal Requirements:

1. Please change "female” or "male" to "woman” or "man" as appropriate, when used as a noun (see for instance https://apastyle.apa.org/style-grammar-guidelines/bias-free-language/gender).

2. Please include a complete copy of PLOS’ questionnaire on inclusivity in global research in your revised manuscript. Our policy for research in this area aims to improve transparency in the reporting of research performed outside of researchers’ own country or community. The policy applies to researchers who have travelled to a different country to conduct research, research with Indigenous populations or their lands, and research on cultural artefacts. The questionnaire can also be requested at the journal’s discretion for any other submissions, even if these conditions are not met.  Please find more information on the policy and a link to download a blank copy of the questionnaire here: https://journals.plos.org/globalpublichealth/s/best-practices-in-research-reporting. Please upload a completed version of your questionnaire as Supporting Information when you resubmit your manuscript.

Additional Editor Comments (if provided):

The three reviewers appointed to conduct peer review of the manuscript have made their respective observations and submitted recommendations that cover the spectrum of weaknesses exhibited in the manuscript. I encourage you to Please kindly read through carefully, they may appear extensive but these are corrections that would strengthen the manuscript for publication in the PLoS Global Public Health Journal.

Reviewers' comments:

Reviewer's Responses to Questions

**Comments to the Author**

1. Does this manuscript meet PLOS Global Public Health’s publication criteria? Is the manuscript technically sound, and do the data support the conclusions? The manuscript must describe methodologically and ethically rigorous research with conclusions that are appropriately drawn based on the data presented.

Reviewer #1: Yes

Reviewer #2: Partly

Reviewer #3: Yes

2. Has the statistical analysis been performed appropriately and rigorously?

Reviewer #1: Yes

Reviewer #2: Yes

Reviewer #3: Yes

3. Have the authors made all data underlying the findings in their manuscript fully available (please refer to the Data Availability Statement at the start of the manuscript PDF file)?

Reviewer #1: Yes

Reviewer #2: Yes

Reviewer #3: Yes

4. Is the manuscript presented in an intelligible fashion and written in standard English?

Reviewer #1: Yes

Reviewer #2: Yes

Reviewer #3: Yes

5. Review Comments to the Author

Reviewer #1: 1.The author should indicate the dates the following references were retrieved: 12, 13, 14, 15, 16,17,18,9, 22,25,26,27,28,29-30,33, 36,37

2. The author should add ''Implication of the study'' as well as ''Contribution to body of Knowledge''

Reviewer #2: Thank you for the opportunity to review your paper entitled “COVID-19 Vaccine Hesitancy and Confidence in the Philippines and Malaysia: A Cross-sectional Study of Sociodemographic Factors and Information-Seeking” which aimed to examine the differences in factors associated with COVID-19 vaccine hesitancy and vaccine confidence in Malaysian vs. Philippines residents. This paper addresses an important topic and has the potential to inform public policy and tailored health interventions.

Below are my recommendations/concerns-

• Vaccine hesitancy and vaccine confidence are related but distinct constructs. The author(s) define vaccine hesitancy, but no definition is provided for vaccine confidence. How was vaccine confidence conceptualized for the purpose of this study? Please, provide a clear definition for vaccine confidence and some information about how both constructs are related.

• The author(s) discussed historical events in the Philippines that may impact vaccine confidence and uptake. Medical mistrust could be an appropriate proxy to assess this theorized phenomenon.

• The introduction would be stronger if the author(s) include a discussion on factors known to be associated with vaccine hesitancy and confidence based on extant literature. Including that background information would help the reader better understand the selection of independent variables, like religion for instance, in the analysis.

• Besides age, what eligibility criteria was used to define the target population? Was language proficiency a factor? Was the survey written in English? After reading the Methods section, it is still unclear whether only unvaccinated Filipino and Malaysian residents were included in the study given that the aim of the study is to assess vaccine hesitancy and confidence levels in “unvaccinated individuals”.

• The “Measures” section could use further development. Due to the lack of contextual information, some of the categorization/coding of variable seem arbitrary and unclear. Was age treated as a continuous variable? If yes, that should be clearly stated. If not, how was it categorized? Religion is classified as “Christian/Buddhism/Muslim/Hinduism”. Is that an exhaustive list of potential religious affiliations in Malaysia and Philippines. Was there an “Other” category on the survey? How did you account for participants who do not practice any form of organized religion? In addition, the income variable is categorized as “sufficient vs. insufficient”. Was this subjectively reported by the participant? Otherwise, what income guidelines were used to determine if the participant’s income was sufficient?

• Relatedly, to provide more context, the author(s) should elaborate on how the variables were coded. Another example- was “social media users” posed as a simple yes/no question on the survey or was the variable coded as yes/no based a survey item that assessed frequency of social media use. Essentially, the author(s) should be clearer about how each variable was coded. Though some of the requested information is present in Table 1, it should also be discussed in the Methods section.

• Author(s) should please provide a brief explanation of the MacArthur Scale of Subjective Social Status within the text of the manuscript.

• Vaccine confidence and hesitancy were assessed with two questions. What is the source of the questions/items? Are the items validated? Are they from existing literature? Is there evidence to ascertain that the two questions/items posed on the survey are valid and reliable assessments of vaccine hesitancy and confidence? Also, how were the responses to these items coded for the analysis?

• Is there a scoring system for the DHLI? How are results from the DHLI interpreted?

• Discussion- Lines 227-228 discusses the economic impact of the pandemic. This statement seems out of place given the focus of the paper.

• Other limitations to this study not discussed in the paper. 1) This was an online survey meaning it is selective for Filipinos and Malaysians with access to internet who may be distinctly different from those who do not have access to the internet. 2) I have concerns about the validity and reliability of the items used to assess vaccine hesitancy and confidence. 3) Not all pertinent independent variables are accounted for (past experiences with vaccines, perception of risk/vulnerability, medical mistrust, vaccine knowledge).

Reviewer #3: TITLE OF THE MANUSCRIPT: COVID-19 Vaccine Hesitancy and Confidence in the Philippines and Malaysia: A Cross-sectional Study of Sociodemographic Factors and Information-Seeking

Manuscript Identification: PGPH-D-22-00972

The manuscript submitted for review and possible consideration for publication covers an important subject of contemporary global conversation and seeks make its contributions.

A) TITLE: This title requires minor modification and reviewer suggests "Sociodemographic Factors and Information-Seeking as predictors of COVID-19 Vaccine Hesitancy and Confidence in the Philippines and Malaysia"

The inclusion of the words "factors" and "predictors" has defined the study design and statistical analysis involved and need not appear in the title. Again, observing that your abstract already has the word "predictors" establishes the coherence of the title and the abstract.

B) ABSTRACT: The abstract has not stated the objective of the study in any form. This can be stated as; "This study sought to investigate the extent to which Sociodemographic Factors and Information-Seeking may predict the level of Confidence in COVID-19 Vaccines and the corresponding Hesitancy to accept vaccination among adult participants selected from population in the Philippines and Malaysia"

No mention of the method of instrument validity and reliability including constructs measured and how. The results reported in the abstract should be carefully articulated to provide clarity of groups being compared among other issues. For example, in line 45, the sentence "Results showed that Filipino (vs. Malaysian) participants indicated higher prevalence of COVID-19 vaccine hesitancy (56.6 vs. 22.9%, p =0.001). This statement does not convey sufficient clarity. If the two populations are being compared, then, the statement should be written in the conventional style; "Results showed that Filipino (56.6%) participants exhibited higher prevalence of COVID-19 vaccine hesitancy than Malaysians (22.9%) in the study (p <0.001)".

Note, the p-value should be p<0.001, this is more conventional than p=0.001. It is observed that levels of confidence in the vaccine and information-seeking was not measured or reported in the abstract. This is an important omission.

Although multivariate analysis was reported with Odds Ratio in the main study, outcomes of such analysis did not feature in the abstract. It would be interesting and revealing to report the ORs (Odds Ratio) for predictors between the two populations from line 52-56.

C) INTRODUCTION: Editorial corrections in Line 69 is needed; "...the world’s population are yet to receive a single dose…". Correction is required because still and yet express the same meaning.

Note that the thesis in this study is centered on the statement in lines 84-86 "Thus, developing a deeper understanding of the factors associated with vaccine hesitancy and confidence will be crucial toward informing locally-tailored health promotion strategies.". Therefore, in this study at this time, it is not just understanding that is so much important but the explanations the study will provide based on appropriate scientific principles derived from epidemiology, behaviour theories responsible for the way the population respond to health threats and public health prevention and control.

The study has three important predictors, "sociodemographic factors", "information-seeking behaviour" and "confidence-also referred to as self-efficacy expectation with its associated outcome expectancy". These are constructs grounded in behaviour theories in the domain of Bandura's social cognitive theory, Information-motivation-behavioural skills model, and health literacy definitive of information typologies, that requires clarifications regarding how these constructs are able to provide the elucidation the study seeks.

There is no theoretical foundation to sustain any argument the study may put up in the absence of these theoretical considerations. The absence of a theoretical framework has undermined the validity of the constructs to be measured in the study and any argument being put up in the global conversation related to COVID-19 vaccines and halting the spread of the virus. For instance, information-seeking has no theoretical characteristics of information and conceptual link to the definitions of information, nor health literacy reviewed.

Therefore, it is very pertinent to have a paragraph somewhere in line 96 before statement of objective to place the theoretical and conceptual clarifications of likely underpinning dynamics in the web of interactions between the various constructs leading to the development of a conceptual framework.

Note that the aim of the study is not clearly well stated, as it is absent in the abstract. Suggested revision: “The main aims of this research were to determine levels of hesitancy and confidence in COVID-19 vaccines among unvaccinated individuals in population of the Philippines and Malaysia, and to identify predictors significantly associated with these outcomes.”. It is worthy of note that the use of the word “influencing” cannot be ascertained in a cross-sectional study design despite the use robust statistical analysis.

D) Methodological approach appears adequately conceived and implemented.

E) Similarly, the results appear adequately presented.

F) The discussion would have benefitted from reiterating the main aim of the study before presenting the implications of the data and concluding.

G) The reference listings: Currency of literature cited is impressive but few online resources do not have dates accessed.

6. PLOS authors have the option to publish the peer review history of their article (what does this mean?). If published, this will include your full peer review and any attached files.

**Do you want your identity to be public for this peer review?** For information about this choice, including consent withdrawal, please see our Privacy Policy.

Reviewer #1: **Yes: **PROF IBRAHEEM SHOLA ABDULRAHEEM, Department of Epidemiology & Community Health, College of Health Sciences, University of Ilorin, Nigeria

Reviewer #2: No

Reviewer #3: **Yes: **Bola Christie ATULOMAH (PhD)

---

## [Editor Report · Decision Letter 1]

26 Sep 2022

COVID-19 Vaccine Hesitancy and Confidence in the Philippines and Malaysia: A Cross-sectional Study of Sociodemographic Factors and Digital Health Literacy

PGPH-D-22-00972R1

Dear Dr. Brackstone,

We are pleased to inform you that your manuscript 'COVID-19 Vaccine Hesitancy and Confidence in the Philippines and Malaysia: A Cross-sectional Study of Sociodemographic Factors and Digital Health Literacy' has been provisionally accepted for publication in PLOS Global Public Health.

Best regards,

Nnodimele Onuigbo Atulomah, PhD

Academic Editor

The authors have indeed worked very hard and meticulously to address every point raised with appropriate responses. congratulations.